# Naturally-derived protein extract from *Gryllus bimaculatus* improves antioxidant properties and promotes osteogenic differentiation of hBMSCs

Keya Ganguly[1☉], Sayan Deb Dutta[1☉], Min-Soo Jeong[2], Dinesh K. Patel[1], Seong-Jun Cho[2]*, Ki-Taek Lim[1]*

1 Department of Biosystems Engineering, Institute of Forest Science, College of Agriculture and Life Sciences, Kangwon National University, Chuncheon, Republic of Korea, 2 Department of Food Science and Biotechnology, College of Agriculture and Life Sciences, Kangwon National University, Chuncheon, Republic of Korea

☉ These authors contributed equally to this work.
* sj.cho@kangwon.ac.kr (S-JC); ktlim@kangwon.ac.kr (K-TL)

**Data Availability Statement:** All relevant data are within the manuscript and its Supporting Information files.

## Abstract

Naturally-derived proteins or peptides are promising biopolymers for tissue engineering applications owing to their health-promoting activity. Herein, we extracted proteins (~90%) from two-spotted cricket (*Gryllus bimaculatus*) and evaluated their osteoinductive potential in human bone marrow-derived mesenchymal stem cells (hBMSCs) under *in vitro* conditions. The extracted protein isolate was analyzed for the amino acid composition and the mass distribution of the constituent peptide fraction. Fourier transform infrared (FTIR) spectroscopy was used to determine the presence of biologically significant functional groups. The cricket protein isolate (CPI) exhibited characteristic protein peaks in the FTIR spectrum. Notably, an enhanced cell viability was observed in the presence of the extracted proteins, showing their biocompatibility. The CPI also exhibited antioxidant properties in a concentration-dependent manner. More significant mineralization was observed in the CPI-treated cells than in the control, suggesting their osteoinductive potential. The upregulation of the osteogenic marker genes (*Runx2*, *ALP*, *OCN*, and *BSP*) in CPI treated media compared with the control supports their osteoinductive nature. Therefore, cricket-derived protein isolates could be used as functional protein isolate for tissue engineering applications, especially for bone regeneration.

## Introduction

Millions of patients suffer from bone-related disorders globally, including osteoporosis, osteosarcoma, and chondrosarcoma, which require bone replacement, and surgical interventions [1]. Autologous and allograft bone replacement techniques are useful clinical procedures for orthopedic treatment; however, bone tissue engineering is an emerging alternative to the existing medical procedures [2]. The efficiency of bone tissue regeneration *in vitro* and *in vivo* is

**Funding:** K.T.L: 'National Research Foundation of Korea' (NRF) funded by the 'Ministry of Education' (NRF-2018R1A6A1A03025582 & NRF-2019R1D1A3A03103828) S.J.C: Innovative Cultured Meat Technology Development Alchemist Project (20012439) funded by the Ministry of Trade, Industry, and Energy (MoTIE, South Korea) The funders had no role in study design, data collection and analysis, decision to publish, or preparation of the manuscript.

**Competing interests:** The authors have declared that no competing interests exist.

augmented by using suitable osteoconductive and osteoinductive biomaterials [3,4] and bioactive molecules [5]. The tissue regeneration process is extensively supported by proteins, such as transforming growth factor-β (TGF- β) superfamily proteins, fibroblast growth factors (FGFs), insulin-like growth factors (IGFs), and platelet-derived growth factors (PDGFs) [6,7]. However, the application of bioactive protein molecules can augment the bone formation rate during the healing process. Thus, it is crucial to identify bioactive proteins that can promote the complex bone formation process under clinical conditions.

Animals and plants are promising sources of numerous osteoinductive compounds [8–10]. Insects also constitute an enriched source of proteins, essential fatty acids, and minerals. *Gryllus bimaculatus*, known as the two-spotted cricket, is extensively utilized in food science research and traditional medicine for its protein content [11]. Moreover, cricket-derived proteins also show antioxidative and immunomodulatory properties [12]. However, the bone tissue regeneration potential of cricket-derived protein material has not yet been fully explored. An *in vitro* assessment of the effect of cricket-derived protein on the osteogenic differentiation of human bone-marrow-derived mesenchymal stem cells (hBMSCs), which are involved in the development and maintenance of osteoblasts, osteocytes, and osteoclasts [13–18], could provide insights into the efficacy of the cricket-derived protein as an osteoinductive biopolymer.

This study aimed to evaluate the effects of the cricket protein isolate (CPI) on the osteogenic differentiation of hBMSCs under *in vitro* conditions for its potential clinical applications. CPI was characterized using Fourier transform infrared (FTIR) spectroscopy and scanning electron microscopy (SEM) to assess the topological-mediated interaction between CPI and hBMSCs. In addition, the protein mass range and amino acid composition were determined to identify the effects of the nutritional composition of CPI on hBMSC osteogenic differentiation. Additionally, the antioxidative properties of CPI were analyzed to evaluate its efficiency in reducing cellular oxidative stress. Moreover, the biocompatibility, mineralization efficiency, and variations in the osteogenic marker gene expression in the presence of CPI were investigated to confirm its possible roles in hBMSC osteoinduction. We anticipate that CPI can be used as an osteoinductive material for the fabrication of osteoinductive scaffolds or as an alternative to expensive protein supplements to treat bone-related diseases.

## Materials and methods

### Materials

Crickets were collected from Wonju Natural Ecology Park (Wonju, Republic of Korea). Laemmli buffer (5×) and protein ladder were purchased from Dyne Bio Inc., Seongnam, Republic of Korea. Coomassie Brilliant Blue and SYBR Green Master mix were supplied by Bio-Rad Laboratories, USA. Dulbecco's Modified Eagle's Medium (DMEM), 10% fetal bovine serum (FBS), Dulbecco's phosphate-buffered saline (DPBS), and antibiotics were purchased from Welgene Inc., Republic of Korea. Trypsin-ethylene diamine tetra acetic acid (Trypsin-EDTA) was provided by Gibco, USA. Osteo-induction media, 4,6-diamino-2-phenylindole dihydrochloride (DAPI), and alizarin red (ARS) staining kit were acquired from Sigma-Aldrich, USA. WST-1 dye and Alexa Fluor conjugated monoclonal antibodies were purchased from DoGenBio Co., Ltd., Republic of Korea, and Santa Cruz Biotechnology, USA, respectively. TRIzol® reagent, Acridine orange, and Ethidium bromide stains were purchased from Invitrogen, Thermo Fisher Scientific, USA. The cDNA synthesis kit was obtained from Invitrogen, Gaithersburg. The gene primers were supplied by BIONEER®Inc., Daejeon, Republic of Korea.

## Defatting and protein extraction

Crickets were fed with corns and cabbages and were harvested at 42 days from their hatching. The killing of crickets was accomplished with $CO_2$ treatment and stored at -80˚C for further experiments. The stored crickets were lyophilized and ground by a pulverizer (RT-N08, Rong Tsong Precision Technology, Taichung, Taiwan). The defatting process was done, as described earlier [19]. Briefly, the required amounts of cricket powders were dispersed in an ethanol solution (99.5%) and agitated for 4 h, and the solvent was replaced within 2 h of intervals. The filtrate was left to dry at room temperature, followed by the addition of the required amounts of water, and stirred for 30 min. The pH of the suspension was maintained to 11 by adding 2.5 M sodium hydroxide (NaOH) and agitated for 60 min at room temperature (RT). The mixture was centrifuged at 3200*g* for 20 min and filtered by Whatman grade 41 filter paper (Whatman plc, Maidstone, UK). The isoelectric point of CPI (pH 4.0) was achieved by adding 2.5 M HCl solution in the supernatant and subsequently agitated for 30 min at RT. After this, the acidified mixture was centrifuged at 3200*g* for 20 min, followed by lyophilization. The dried sample was stored till further use. The schematic representation for the extraction of protein from the insect is given in **Fig 1**.

## CPI chemical characterization

**Determination of the nutrient composition.** The amino acid content in the CPI was analyzed using an amino acid analyzer (HITACHI L-8900, Hitachi High-Technologies

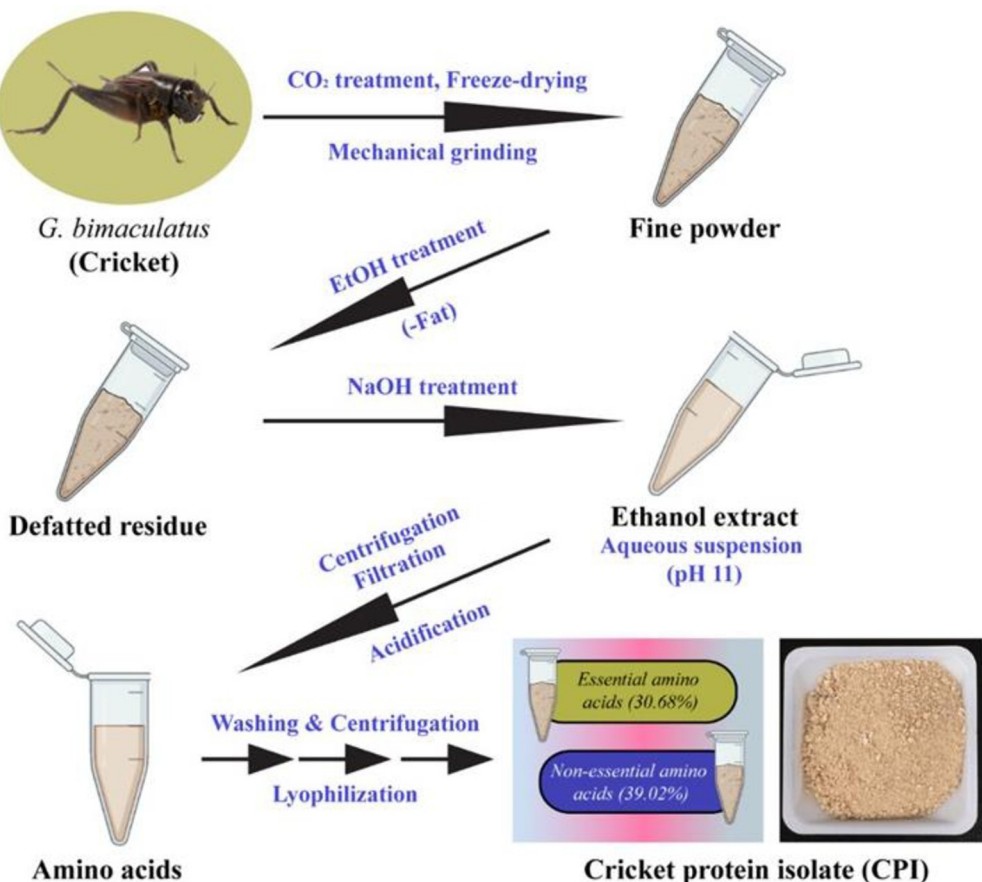

**Fig 1. Schematic illustration of protein extraction from *Gryllus bimaculatus*.**

Corporation, Japan) with the ninhydrin method. The proximate composition of the CPI was analyzed according to the AOAC methods [20]. The protein content of CPI was determined using the Kjeldahl method with a protein-to-nitrogen conversion factor of 6.25. The ash content was measured by burning the sample in a furnace. The crude fat was analyzed according to the Soxhlet method measuring the crude fat extracted by ether. The carbohydrate content of the sample was calculated as follows, *Carbohydrate content (%) = 100 – [Crude protein (%, DM) + Crude fat (%, DM) + Crude ash]*, where DM represents dry matter.

**Molecular weight determination.** The molecular weight distribution of CPI was analyzed by a modified sodium dodecyl sulfate-polyacrylamide gel electrophoresis (SDS-PAGE) as described previously [21]. Briefly, the samples were dissolved in 8 M urea solution and stirred for 30 min at 25°C, followed by centrifugation at 3200$g$ for 20 min. An aliquot of the supernatant was added to a 5× sample buffer containing 62.5 mM $\beta$-mercaptoethanol, 10% SDS, and 0.1% bromophenol blue. The proteins were denatured at 95°C for 4 min, followed by the electrophoresis of 4 μL of protein ladder (12–160 kDa) and 10 μL of the denatured sample, and separated by 10% Tris-Glycine gel. After the electrophoresis, the gel was stained with 0.05% Coomassie Brilliant Blue and destained with a destaining solution (acetic acid: methanol: deionized water = 2: 5: 5).

MALDI-TOF spectral analysis was done for the water-soluble fraction of the CPI after trypsin digestion of CPI at 37°C for overnight using a Mass Spectrometer (Bruker Autoflex TOF/TOF, Bruker, Germany) within 1000–6000 m/z resolution.

**Chemical composition and morphological analysis.** The Perkin Elmer FTIR analyzer (Frontier, Perkin Elmer, UK) was applied to evaluate the functional groups present in the sample in a transmitted mode in the wavenumber range of 4000–1000 cm$^{-1}$ with a resolution of 4 cm$^{-1}$. The morphology of the sample was monitored by high-resolution field emission-scanning electron microscopy (FE-SEM) (S-4800, Tokyo, Japan).

## DPPH assay

The radical scavenging activity was determined by the DPPH assay. Briefly, different CPI concentrations (0.25%, 0.5%, 1% and 2%) were prepared in 0.4 mM DPPH solution. The mixture was incubated for 30 min in dark conditions at RT. The ascorbate solutions (0.25%, 0.5%, 1%, and 2%) were also prepared for reference. The optical density (OD) was recorded at 517 nm using a spectrophotometer (Infinite® M Nano 200 Pro; TECAN, Switzerland). The OD values were plotted to compare the DPPH scavenging activity of the CPI with respect to the equivalent concentration of ascorbate.

## Cell culture

The hBMSCs were received from the Korean Cell Line Bank (KCLB, Seoul, Republic of Korea) and cultured as previously reported [22,23]. The cell culture was carried out using DMEM supplemented with 10% FBS and 1% antibiotics containing penicillin (10,000 units/mL), streptomycin (10,000 μg/mL), and amphotericin B (25 μg/mL) at 37°C in a humidified atmosphere of 5% $CO_2$ (Steri-Cycle 370 Incubator; Thermo Fisher Scientific, USA). The old media were changed with fresh media every three days during the experiment. After 70–80% confluency, the hBMSCs were treated with different concentrations of CPI for the desired periods. Passage 5 cells were used in this study. For osteogenic induction, the cells were cultured in an osteogenic induction media containing DMEM supplemented with 50 μg/mL L-ascorbic acid, 10 mM β-glycerophosphate, and 100 nM dexamethasone.

## Cell viability assay

The hBMSCs ($1 \times 10^4$ cells/100 μL media) were seeded into a 96-well plate and incubated with 0.25%, 0.5%, 1%, and 2% CPI at 37˚C with 5% $CO_2$ for the chosen periods (1, 3, and 5 days) after 70–80% of confluency. The media without the treatment were considered as control. The cell viability was analyzed using WST-1 assay (EZ-Cytox Cell Viability Assay Kit®). After the treatment, 10 μL of the WST-1 dye was added and further incubated for 2 h. The produced formazan was quantitated by measuring the absorbance at 450 nm (625 nm as a reference value). All the experiments were accomplished in triplicate, and data are presented as mean ODs ± standard deviations. Statistical significance was considered at $p^* < 0.05$.

## Live-dead assay

For this, the hBMSCs ($4 \times 10^4$ cells/100 μL media) were cultured in a 4-well plate at 37˚C with 5% $CO_2$, followed by 2% CPI treatment after 70–80% of confluency. The cells grown in DMEM alone were taken as control. The cells were washed with 1× PBS, followed by treatment with 1μL of acridine orange and ethidium bromide dye solution at a ratio of 1:1. The images were captured immediately on appropriate filter channels using Leica Microsystems Suite X software (Leica Microsystems, Germany) of the inverted fluorescence microscope (DMi8 Series, Leica Microsystems, Germany). The survivability of the CPI treated cells was quantified using the live-dead fluorescence imaging after 3 days of incubation.

## Colony formation and morphological analysis

The colony formation efficiency and morphology of hBMSCs in the presence of CPI was investigated using (1) Giemsa staining, (2) bright-field microscopy, and (3) fluorescence microscopy.

For Giemsa staining, hBMSCs ($4 \times 10^4$ cells/100μL media) were cultured in the presence of different concentrations of CPI and incubated for 3 days. After 80% confluency, the cells were washed with PBS. The washed cells were fixed with 3.7% PFA at RT. The media without treatment and gelatin were considered as negative and positive control, respectively. The fixed cells were washed with PBS and permeabilized with 100% methanol for 20 min. The permeabilized cells were further washed with PBS and incubated for 10 min with Giemsa stain. The excess stain was removed by washing with PBS, and images were captured at the magnification of 5× under an optical microscope (Zeiss Optical Microscope, USA).

The arrangement of F-actin was studied through fluorescence imaging to visualize the effect of 2% CPI on the cell cytoskeleton. The hBMSCs ($2 \times 10^4$ cells/100 μL media) were cultured in 60 mm bottom well plates and treated with 2% CPI for 3 days. The media without the CPI was taken as control. The staining of cells was performed as described earlier with some modifications [23]. Briefly, the cells were washed with PBS and fixed with 3.7% paraformaldehyde (PFA) for 15 min at room temperature, followed by the addition of 0.1% Triton X-100 to permeabilize the cells for 10 min at RT. The cells were rinsed twice with PBS buffer and blocked for 1 h with 1% BSA. The permeabilized cells were rinsed with PBS and then incubated for 30 min with 200 μL Alexa Fluor (AF) 488 F-actin probe (ex/em = 488/518) to visualize the F-actin. The nuclear staining was done with the addition of 20 μL of 1 mg/mL DAPI solution for 2 min in the dark. The stained cells were rinsed and covered with a mounting medium and a glass coverslip. The fluorescence images were taken with a fluorescence microscope at a magnification of 40×. The ROI intensity of the images was quantified using ImageJ software (ImageJ v1.8, NIH Lab., USA, www.imagej.nih.gov).

## Mineralization study

The effect of CPI on the mineralization of hBMSCs was evaluated by the ARS procedure after 7, 14, and 21 days of treatment. The used media were replaced with fresh media every 3 days. The cultured cells were rinsed with PBS. The cells were fixed and permeabilized with 1 mL of 70% absolute ice-cold ethanol for 15 min at RT. The permeabilized cells were stained with 500 μL of 40 mM ARS (pH 4.2) stain for 10 min, followed by washing with deionized water to remove the excess stain. The mineralization was documented using the optical microscope. The mineral quantification was estimated by dissolving the formed mineral in 500 μL destaining solution (10% cetyl pyridinium chloride and 10 mM sodium phosphate). The absorbance of the solution was taken at 562 nm using a spectrophotometer. All the samples were prepared in triplicate, and data are presented as mean ODs ± standard deviations. Statistical significance was considered at $p^* < 0.05$.

## ALP activity

The ALP activity of CPI-treated hBMSCs was evaluated by immunohistochemical (IHC) staining methods as described previously [24]. The cells were fixed with 3.7% paraformaldehyde (PFA) for 30 min at RT, followed by permeabilization with 0.1% Triton X-100 for 5 min. The fixed and permeabilized cells were blocked with 1% BSA for 1h, followed by incubation with mouse monoclonal antibody against ALP. After that, the cells were washed twice by PBS and reacted with 30 μL of DUB substrate kit (ab64238, Abcam, USA) and incubated for 10–15 min at RT. Next, the plates were rinsed twice with PBS and reacted with 5 μL of 30% $H_2O_2$ to block endogenous peroxidase activity. The reaction proceeded until the desired color was achieved. Finally, the plates were washed twice with PBS and counterstained with hematoxylin stain (Sigma-Aldrich, USA) and visualized by an inverted optical microscope. Images were captured with 20× magnification and compared with control groups. A score of 0 (negative), 2–3 (mild), 4–8 (moderate), and 9–12 (strongly positive) was assigned to the semiquantitative evaluation of the IHC reaction for control and CPI-treated groups. The final IHC score was represented in terms of a 0–12 score.

## RNA isolation and real-time PCR (qRT-PCR) analysis

The expression of the osteogenic-marker genes in CPI treated and control cells were evaluated by the qRT-PCR technique. Briefly, the cells ($4 \times 10^4$ cells/100μL media) were cultured in a 24-well plate in the osteogenic induction media for 7 and 14 days, followed by the extraction of RNA by TRIzol® reagent (Thermo Fisher Scientific, USA), according to the manufacturer's instructions. The purity and concentration of the extracted RNA were evaluated by a spectrophotometer. The cDNA was synthesized from 2 μg of RNA by using reverse transcriptase and SYBR Green Master mix. The mRNA expression was quantified with a Bio-Rad Real-Time PCR (CFX96TM Maestro Real-Time System, Bio-Rad, USA). The reaction condition included 43 cycles of denaturation for 15 sec at 95˚C and 1 min amplification at 60˚C. All the experiments were performed in triplicate and normalized to the housekeeping gene β-actin. The relative mRNA expression from hBMSCs in the presence of CPI and control was compared in a histogram. All the samples were prepared in triplicate during the experiments. The specific gene primers used for qRT-PCR analysis are listed in **Table 1**.

## Immunocytochemical staining

The expression of the osteogenic marker proteins was studied through immunocytochemical staining procedure. The hBMSCs ($4 \times 10^4$ cells/100 μL media) were cultured in 60 mm bottom well plates and treated with 2% CPI for 7 and 14 days. The media without the CPI was taken as control. The staining of cells was performed by washing with PBS, followed by fixing with 3.7%

**Table 1. Specified gene primer sequences used in qRT-PCR analysis.**

| Genes | *GenBank* Accession No. | Sequences (5′ to 3′) |
|---|---|---|
| *β-actin* | NM_031144 | ACCCGCGAGTACAACCTTCT<br>CTTCTGACCCATACCCACCA |
| *Runx2* | NM_001146038 | CGCACGACAACCGCACCAT<br>CAGCACGGAGCACAGGAAGTT |
| *OSX* | NM_001300837 | TGCTTGAGGAGGAAGTTCAC<br>AGGTCACTGCCCACAGAGTA |
| *ALP* | NM_007431 | CCAACTCTTTTGTGCCAGAGA<br>GGCTACATTGGTGTTGAGCTTTT |
| *BSP* | L09555 | AACTTTTATGTCCCCCGTTGA<br>TGGACTGGAAACCGTTTCAGA |
| *OCN* | AL135927 | TGAGAGCCCTCACACTCCTC<br>ACCTTTGCTGGACTCTGCAC |
| *OPN* | J04765 | TGAAACGAGTCAGCTGGATG<br>TGAAATTCATGGCTGTGGAA |
| *COL1* | NM007742 | GCTCCTCTTAGGGGCCACT<br>CCACGTCTCACCATTGGGG |

**Abbreviations:** *β-actin*; Actin beta, *Runx2*; Runt-related transcription factor-x2, *OSX*; Osterix, *ALP*; Alkaline phosphatase, *BSP*; Bone sialoprotein, *OCN*; Osteocalcin, *OPN*; Osteopontin, and *COL1*; Collagen type-1.

PFA for 15 min at RT. Next, the cells were permeabilized by the addition of 0.1% Triton X-100 for 10 min at RT. After that, the cells were rinsed twice with PBS, blocked by 1% BSA, and incubated with 250 μL of mouse monoclonal antibodies against Runx2, ALP, OCN, and OPN. The specific antibody dilutions are listed in **S1 Table**. The nucleus was counterstained with 20 μL of 1 mg/mL DAPI solution for 2 min in the dark. The stained cells were rinsed and covered with a mounting medium and a glass coverslip. The fluorescence images were taken with a fluorescence microscope at a magnification of 40×. The mean fluorescence intensity of the images was quantified using ImageJ software (ImageJ v1.8, NIH Lab., USA, www.imagej.nih.gov).

## Statistical analysis

Statistical analysis was performed using OriginPro 9.0 software. Statistical significance between the control and treatment groups was determined using one-way ANOVA. All the data are presented as mean ± SDs. Differences were considered significant at $^*p < 0.05$.

## Results

### Nutritional composition

The nutritional composition of the CPI was determined, and the values are presented in **Table 2**. The obtained data indicate that the extracted material was enriched in protein content (~ 90.91%). The carbohydrate, ash, and fat content were negligible.

**Table 2. Composition (g/100g) of *Gryllus bimaculatus* protein isolate (mean ± S.D., n = 3).**

| Sample | Crude protein (%, DM) | Crude fat (% DM) | Crude ash (%, DM) | Carbohydrate* (%, DM) |
|---|---|---|---|---|
| Cricket protein isolate | 90.91±0.36[a] | 0.46±0.10[a] | 3.67±0.25[a] | 4.96 |

Different letters in each column show significant difference (p < 0.05) between means. *Carbohydrate (%): 100 –crude protein–crude fat–crude ash. DM; dry matter.

**Table 3. Amino acid composition (g/100 g sample) for the protein isolate of *G. bimaculatus*.**

| Essential amino acids | Contents (g/100g sample) | Non-essential Amino acid | Contents (g/100g sample) |
| --- | --- | --- | --- |
| Isoleucine | 3.89 | Aspartic acid | 8.44 |
| Leucine | 6.40 | Serine | 4.15 |
| Lysine | 5.02 | Glutamic acid | 9.15 |
| Methionine | 1.24 | Proline | 3.58 |
| Phenylalanine | 3.60 | Glycine | 3.34 |
| Tyrosine | 3.81 | Alanine | 4.03 |
| Threonine | 3.56 | Cysteine | 1.00 |
| Valine | 4.27 | Arginine | 5.32 |
| Histidine | 1.72 | Non-essential A.A | 39.02 |
| Tryptophan | 1.06 | | |
| Essential A.A | 30.68 | | |

The total amino acid amounts in CPI were determined using the Kjeldahl method and are listed in **Table 3**.

The total amount of amino acids in the isolate was 69.70 g/100 g of sample. The data showed that the CPI contained 30.68% of essential and 39.02% of non-essential amino acids. Leucine was the most commonly found essential amino acid (6.40 g/100 g of sample), followed by lysine and valine (5.02 and 4.27 g/100 g of sample, respectively). Among the non-essential amino acids, glutamic acid (9.15 g/100 g of sample) and aspartic acid (8.44 g/100 g of sample), per 100g CPI, were the most abundant.

## Molecular weight analysis

SDS-PAGE analysis was performed to determine the molecular weight of the proteins present in the CPI, and the results are shown in **Fig 2A**. Lane 1 indicates the protein ladder, ranging

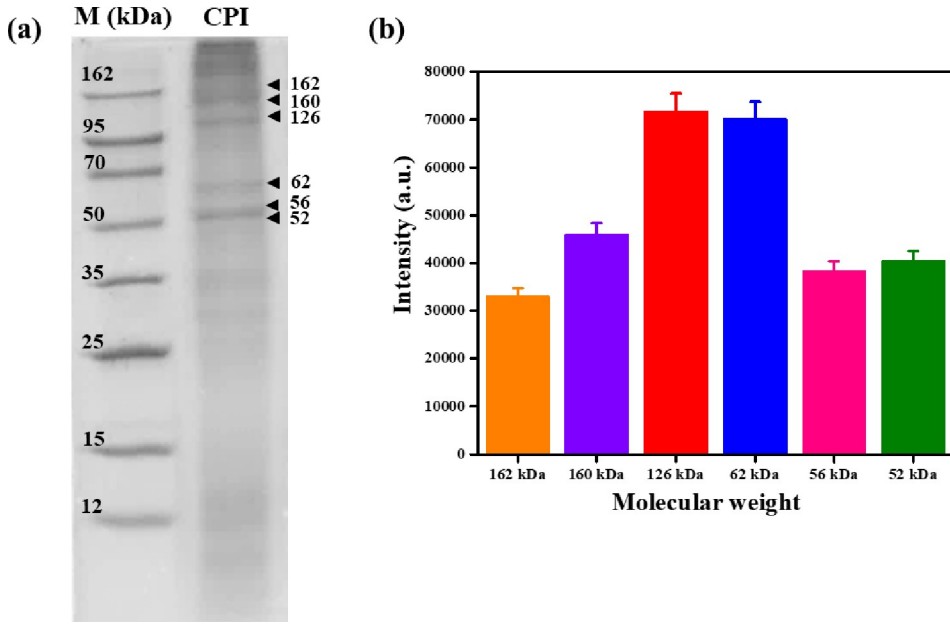

**Fig 2. Determination of molecular weight of CPI. (a)** SDS-PAGE analysis of total CPI, **(b)** SDS-PAGE band intensity profile of total CPI.

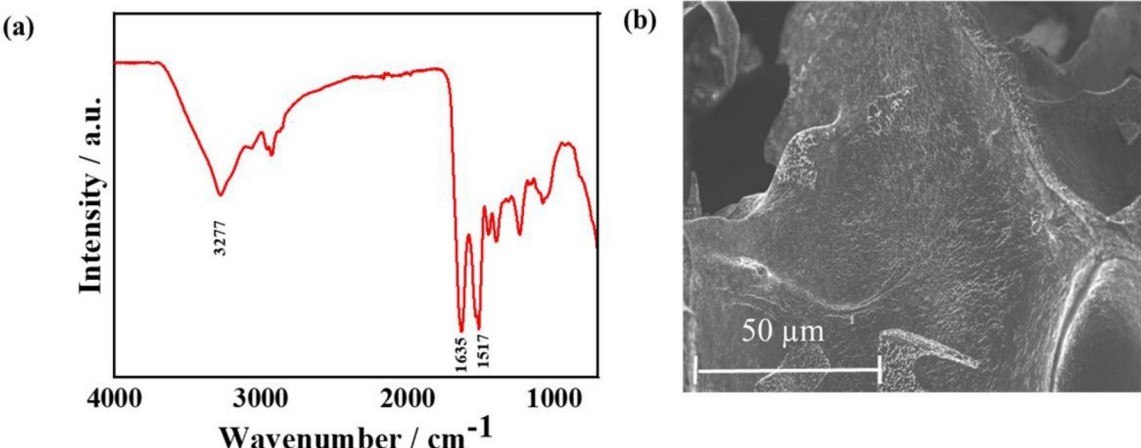

**Fig 3. Chemical characterization of CPI. (a)** The FTIR spectrum of the CPI extracted from insect, and **(b)** FE-SEM morphology of freeze-dried CPI.

from 12 to 160 kDa, while lane 2 shows the CPI protein composition, including six prominent protein bands and a smear of several other faint bands. The protein band intensity profile is shown in **Fig 2B**. The protein fraction of 126 kDa exhibited the highest band intensity, followed by those corresponding to a molecular weight of 62, 160, 52, 56, and 162 kDa. MALDI--TOF MS was performed to determine the molecular weight of proteins present below the range of 6 kDa, and the corresponding mass spectrum is shown in **S1 Fig**. Approximately 26 peaks were observed in the mass spectrum, indicating that different kinds of peptides were found in CPI in the range of 6 kDa.

## FTIR and the morphology of CPI

The FTIR spectrum shows typical protein absorption peaks in the range of 4,000–400 cm$^{-1}$ [25]. The FTIR spectrum of the CPI is shown in **Fig 3A**. The appearance of the absorption peaks at 1,635 and 1,517 cm$^{-1}$ indicates the presence of–C = O (carbonyl) and–N–H (amide II) groups belonging to the $\beta$-sheet structure of proteins [26]. The FTIR absorption peak at 3,277 cm$^{-1}$ indicates the presence of–OH (hydroxyl) or–NH$_2$ (amine) protein groups.

The surface topographical features, including the surface pattern and roughness of the material, play a significant role in cell survivability and osteogenic differentiation. The surface morphology of the freeze-dried CPI was analyzed using FE-SEM, and the micrographs are shown in **Fig 3B**. The CPI exhibited a morphological combination of a rough and smooth layer of flakes, suggesting crystalline structure. However, the surface morphology is profoundly affected by the extraction process and its conditions. Zeta potential measurements were performed to measure the surface potentials of the CPI in water. The colloidal solution exhibited a zeta potential value of -23.2 ± 3.72 mV, indicating that the CPI suspension was electrically stabilized.

## Antioxidant activity of the CPI

The antioxidant potential of CPI was determined using a DPPH assay in the presence of different concentrations of CPI, and the results are presented in **Fig 4A**. A decrease in the absorbance at 517 nm was observed in the CPI-treated conditions compared to the control, showing their scavenging property. This property is highly affected by the CPI concentration, and

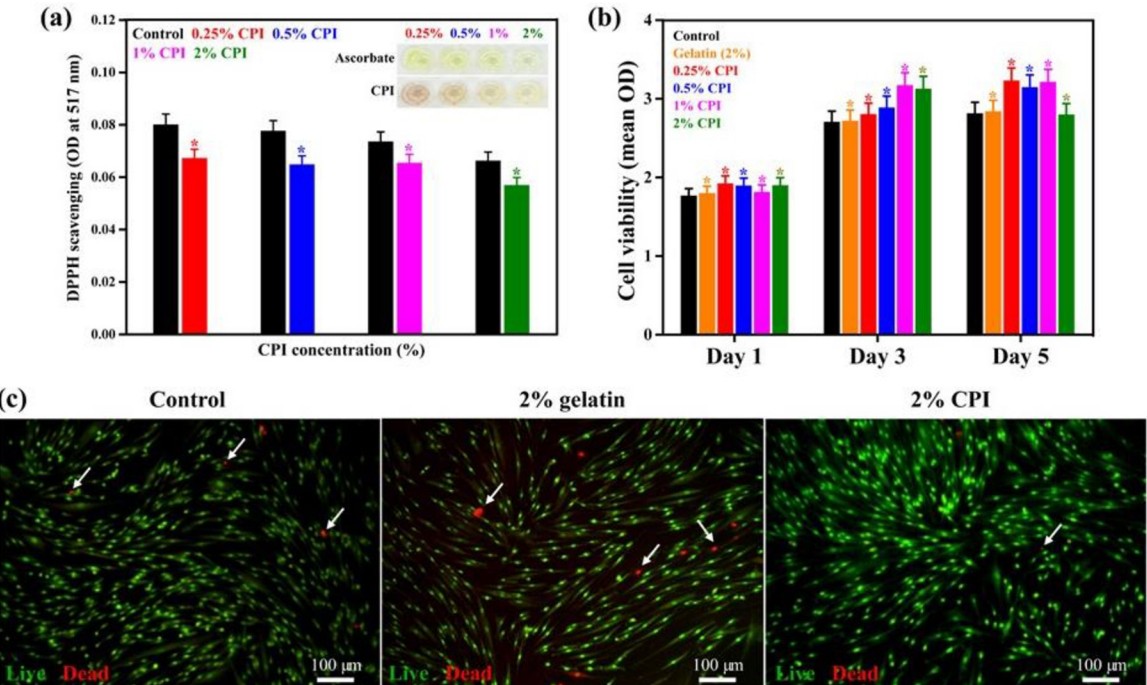

**Fig 4. Antioxidative property and biocompatibility of CPI. (a)** DPPH scavenging activity of CPI in reference to an equivalent concentration of ascorbate with corresponding digital photographs of the plates (inset). **(b)** *In vitro* cytotoxicity evaluation WST-1 assay of CPI-treated hBMSCs. **(c)** Live/dead assay of 2% CPI-treated hBMSCs after 3 days of incubation (Scale bar: 100 μm). Gelatin-treated plates were considered as positive controls. Data are mean ± SD of triplicated experiments, statistical significance at $^*p < 0.05$.

among the concentrations used (0.25%, 0.5%, 1%, and 2%), 2% CPI demonstrated the best scavenging potential as indicated in the digital photographs. A more transparent solution was observed as the concentrations of CPI increased, showing their better scavenging potential. At higher concentrations (2%), the transparency of the CPI solution was similar to that of ascorbate.

## Cell viability and morphology

The cytotoxicity of the CPI was monitored using a WST-1 assay in the presence of hBMSCs, and the results are presented in **Fig 4B**. No adverse effects were observed in the CPI-treated groups, showing their biocompatibility. However, it was interesting to see that the CPI concentrations significantly altered the cell viability, and 2% CPI-treated media resulted in a higher cell viability after 5 days of treatment, indicating a suitable dose for cellular activity. The live-dead assay in the presence of 2% CPI after 3 days of incubation is shown in **Fig 4C**. The mean fluorescence intensities are given in **S2 Fig**. The number of cells that adhered to the substrate was estimated by counting the number of nuclei. Cell death (%) was significantly lower in the CPI-treated group than in the control group, further showing the greater biocompatibility of CPI-treated media.

The colony formation efficiency of hBMSCs in the presence of CPI was monitored using Giemsa staining after 3 days of incubation, and images are shown in **Fig 5A**. The media without CPI and with gelatin were considered as negative and positive controls, respectively. A similar colony formation pattern was observed in the CPI-treated group and in the control group, showing the biocompatibility of CPI-treated media. The cytoskeletal and nuclear

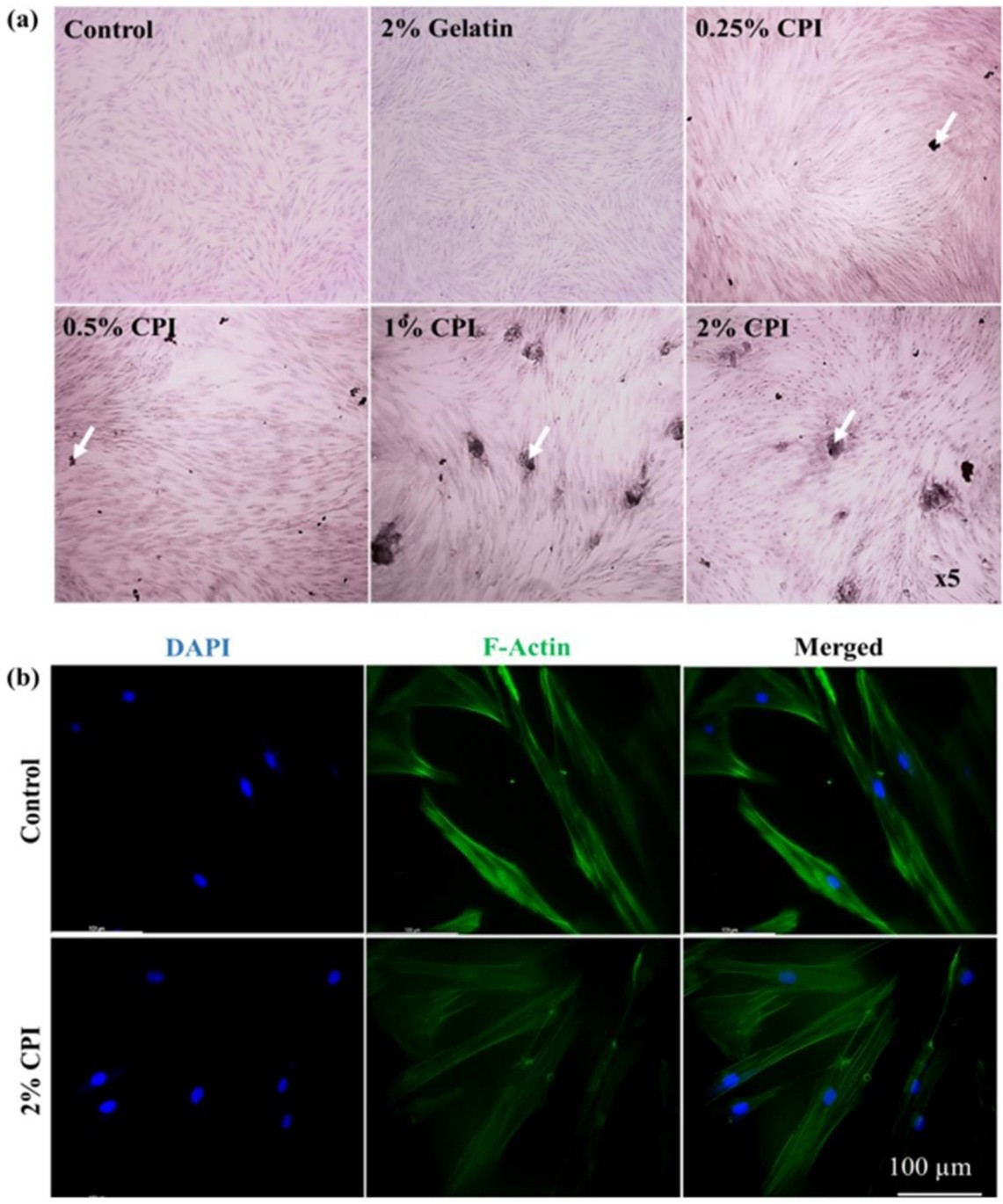

**Fig 5. *In vitro* cytotoxicity evaluation of CPI on hBMSCs after 3 days of incubation. (a)** Colony formation efficiency of hBMSCs using Giemsa staining (Mag. ×5) **(b)** Fluorescence microscopy images of F-actin. The white arrows indicate the presence of stained CPI on the surface of hBMSCs (Scale bar: 100 μm).

morphology of hBMSCs in the presence of CPI was determined using bright-field and fluorescence staining, and the results are shown in S3 Fig. and Fig 5B. Media without CPI were used as a control. It is interesting to note that the crude CPI aggregated on the surface of cells favoring their growth, as imaged by bright-field microscopy. The actin and nuclear morphology of hBMSCs were comparable in CPI treatment and the control conditions, showing no noticeable

cytoskeletal or nuclear damage in CPI-treated cells. The fluorescence intensity profiles of hBMSCs in both conditions are shown in S4 Fig.

## Osteogenic differentiation

The mineralization potential of hBMSCs in the presence of CPI was determined using ARS staining procedures after 7, 14, and 21 days of the treatment, and the results are shown in Fig 6A. An increased mineralization was observed in CPI-treated cells compared with the control, after 7 days of treatment, indicating their mineralization potential. Notably, this trend was further increased after 14 and 21 days of treatment. It is interesting to note that the mineralization potential was strongly affected by the concentration of CPI in the media, and among the concentrations tested (0.25%, 0.5%, 1%, and 2%), 2% CPI exhibited the best mineral deposition potential. The quantitative values of the mineralization are shown in Fig 6B. The higher quantitative values confirm the superior mineralization potential of CPI.

The ALP activity in the presence of CPI was evaluated by IHC following 7, 14, and 21 days of treatment and is represented in Fig 7A. Scores describing the changes in IHC parameters

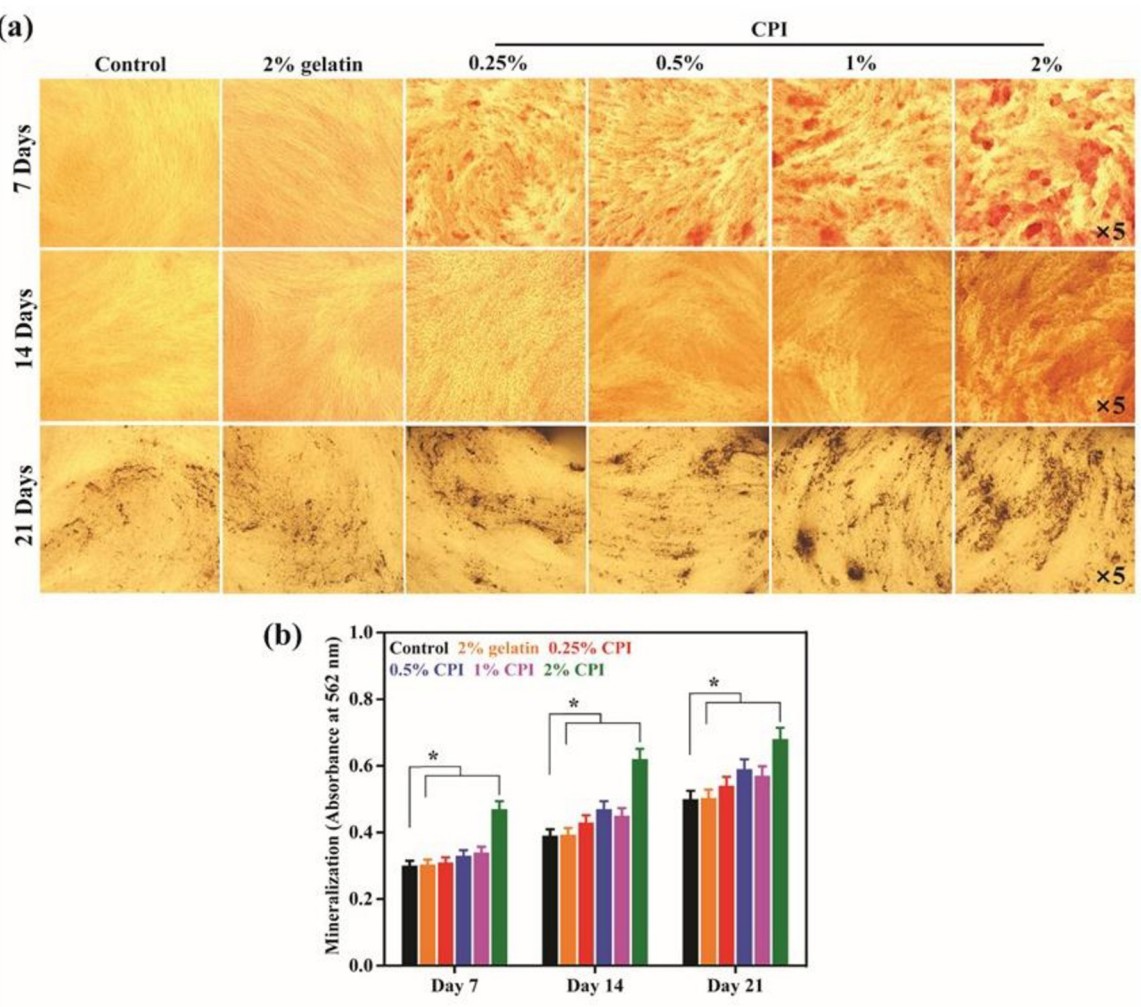

**Fig 6. Evaluation of *in vitro* osteoblast differentiation in the presence of CPI. (a)** ARS staining of CPI-treated hBMSCs after 7 and 14 days of incubation, and **(b)** corresponding quantitative values at indicated time intervals. Data are mean ± SD of triplicated experiments, statistical significance at $^*p<0.05$. The black arrow indicates the presence of stained CPI on the surface of hBMSCs.

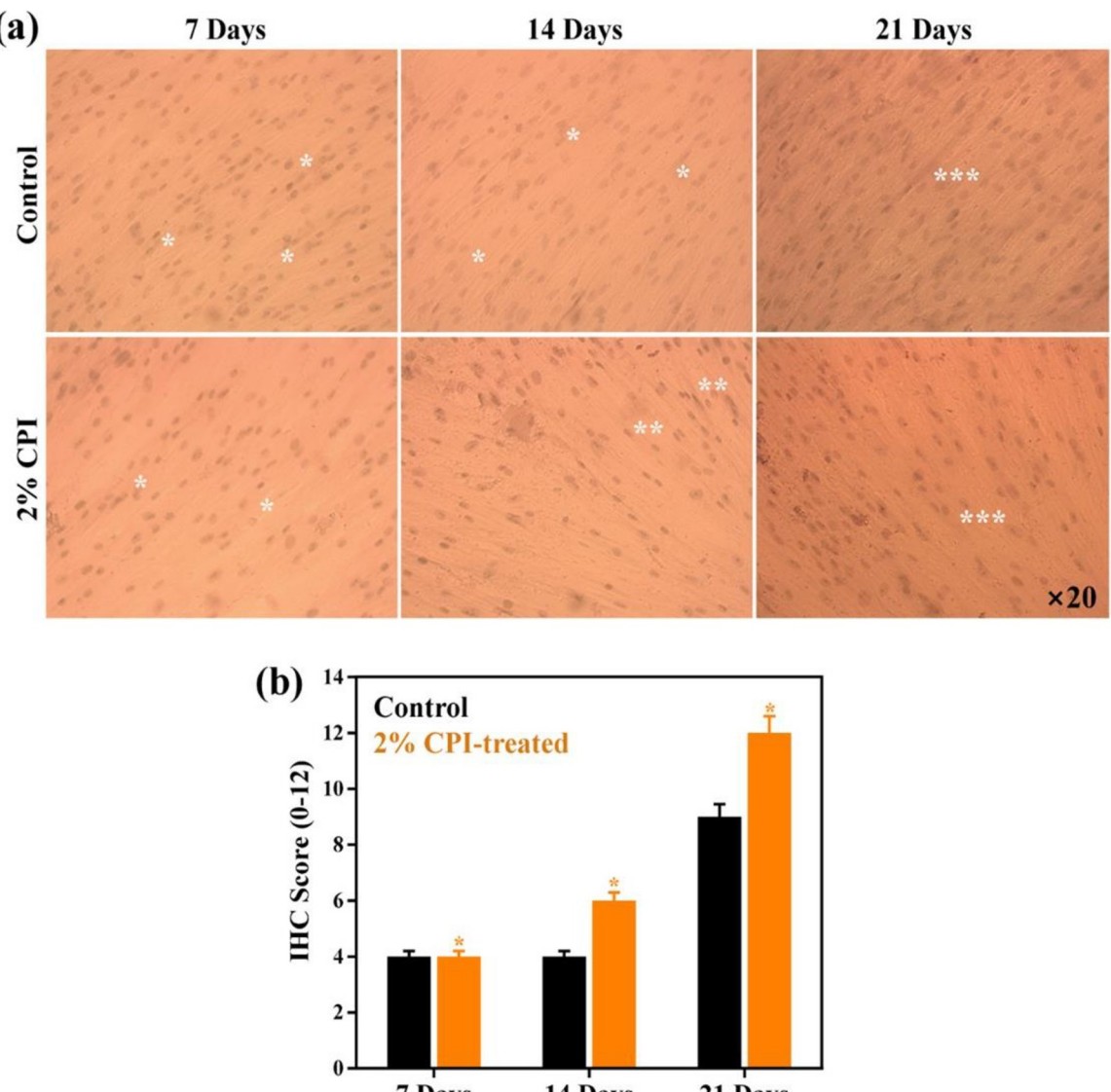

**Fig 7. ALP activity in the presence of CPI. (a)** Representative immunohistochemical (IHC) staining for alkaline phosphatase (ALP) on hBMSCs in control and CPI-treated medium at indicated time intervals. Positive staining is indicated by red-brown cytoplasm followed by hematoxylin counter staining of nucleus. Asterisks with '\*' indicates mild reaction, '\*\*' indicates moderate reaction, and '\*\*\*' indicates intense reaction, **(b)** their corresponding IHC scores. Statistical significance at $^*p < 0.05$.

(percentage of positive cells and staining intensity) is given in **Fig 7B**. The results indicate an increase in the IHC score for the CPI-treated group, meaning that CPI-treated cells exhibited a strongly positive reaction for ALP after 14 and 21 days of incubation. However, a lower expression profile was noticed in control samples than CPI-treated groups suggesting that ALP expression was profoundly affected by CPI-treatment.

## Osteoblast-specific gene and protein marker expression

The expression of osteogenic marker genes (*Runx2*, *OSX*, *ALP*, *BSP*, *OCN*, *OPN*, and *COL1*) in hBMSCs in the presence of 2% CPI and the control after 7 and 14 days of treatment are shown in **Fig 8**. The expression of the early gene marker, *Runx2*, was found significantly higher

Protein biomaterials for improved antioxidant and stem cell differentiation

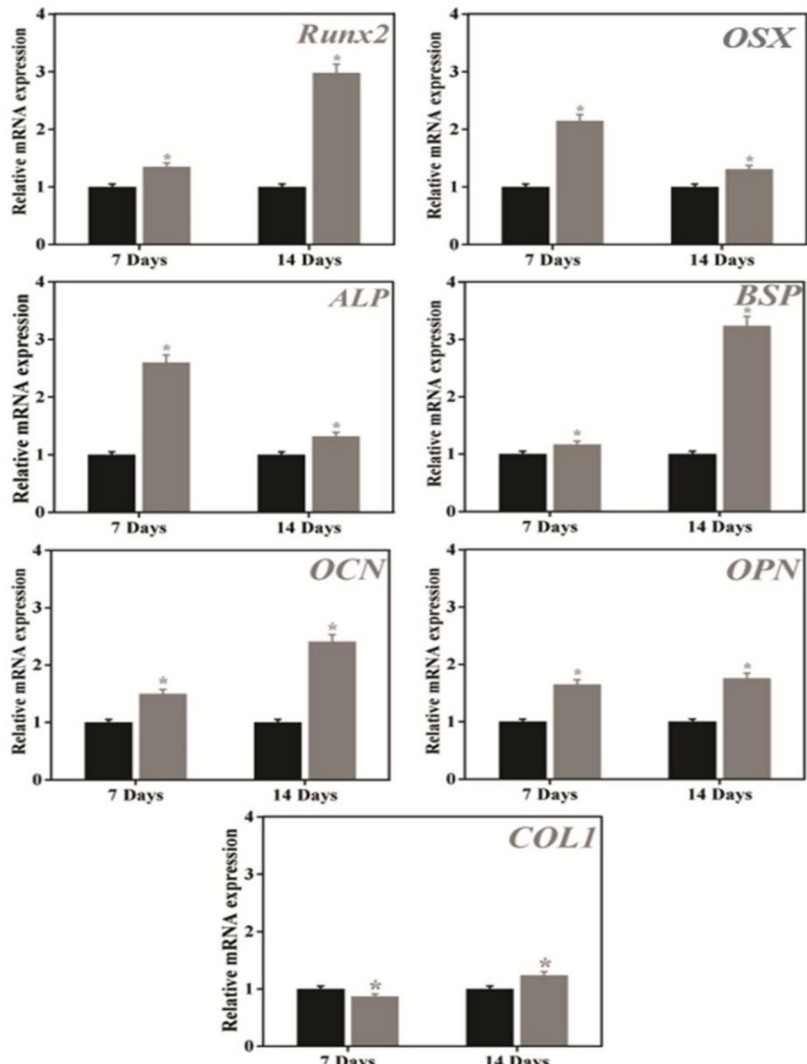

**Fig 8. Real-time polymerase chain reaction (qRT-PCR) analysis of 2% CPI-treated hBMSCs after 7 days and 14 days of incubation.** (a) *Runx2*, (b) *OSX*, (c) *ALP*, (d) *BSP*, (e) *OCN*, (f) *OPN*, (g) *COL1*. Data are mean ± SD of triplicated experiments, statistical significance at *$p < 0.05$.

compared to the control after 14 days of incubation as shown in **Fig 8A**. Besides, the *ALP* expression was significantly increased following CPI treatment, as indicated in **Fig 8B and 8C**. This observation is also supported by the IHC analysis of ALP. In contrast, the expression of *OSX* was decreased after CPI treatment. Interestingly, the expression of *BSP* and *OCN* was also found higher after 14 days of incubation, as shown in **Fig 8D & 8E**. A consistent expression of *OPN* and *COL1* was observed after 7 and 14 days of CPI treatment **Fig 8F & 8G**.

The protein expression of Runx2, ALP, OCN, and OPN is shown in **Fig 9A**. The mean fluorescence intensities for each protein are represented in **Fig 9B**. It was observed that the respective proteins were equally expressed in both controls and CPI-treated cells after 7 and 14 days of treatment; however, no significant differences (*$p < 0.05$) were noted in the expression of Runx2, OCN, and OPN, suggesting that CPI-treatment does not alter their expression even after 14 days of incubation. Besides, ALP expression in CPI-treated cells was significantly higher (**$p < 0.01$) compared to the control groups.

PLOS ONE | https://doi.org/10.1371/journal.pone.0249291   June 2, 2021

14 / 19

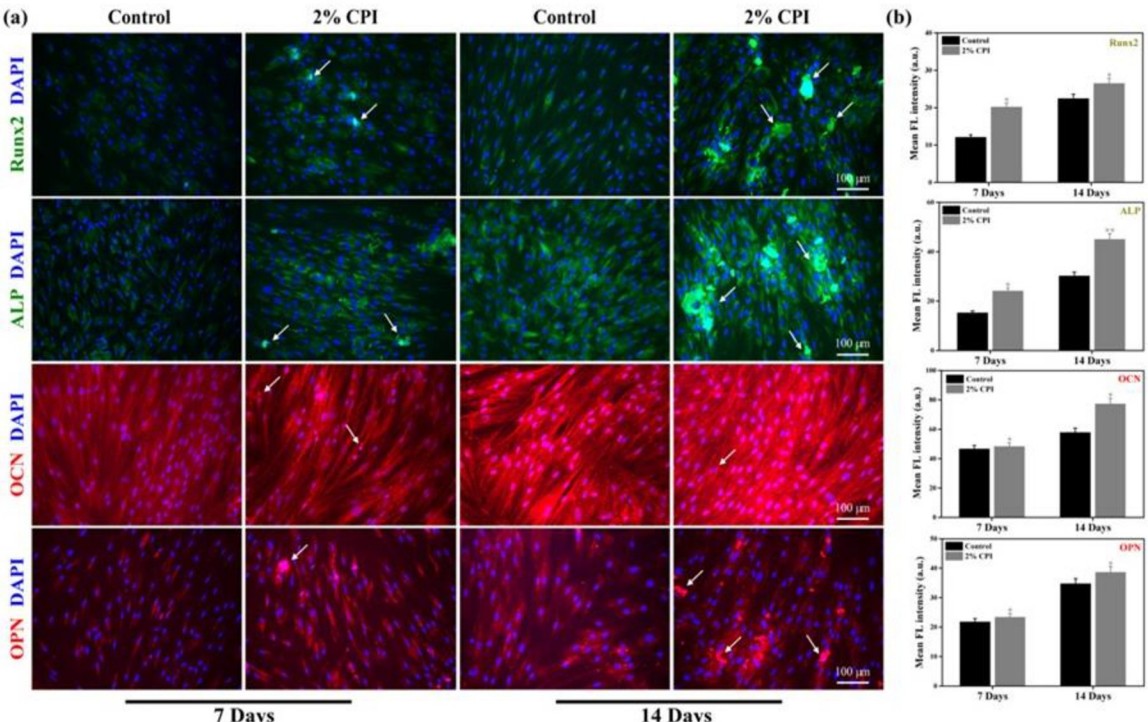

**Fig 9. Osteoblast-specific protein markers expression of CPI treated hBMSCs at indicated time intervals. (a)** Fluorescence microscopy images of respective protein markers (scale bar: 100 μm), and **(b)** corresponding fluorescence intensities. Data are mean ± SD of triplicated experiments, statistical significance at $^*p < 0.05$ and $^{**}p < 0.01$, respectively. White arrow indicates the presence of CPI on to the surface of hBMSCs.

## Discussion

The present study aimed to investigate the osteoinductive potential of a protein isolate extracted from field crickets (*Gryllus bimaculatus*) on the osteogenic differentiation of hBMSCs under *in-vitro* conditions. Since crickets are edible insects [27], their nutritional value is well known [28]; however, they have not been explored thoroughly for their tissue engineering applications. Our results show a successful extraction of ~ 90% of CPI from dried cricket samples, which is substantially higher than a few commercially available cricket-derived protein isolates (~ 45%– 70%) [29,30]. The abundance of glutamic acid in our extracted CPI (9.15 g/100 g sample) is expected to be a crucial factor for promoting osteogenesis. Glutamine has long been recognized as a vital amino acid for stem cell differentiation into osteoblasts [31] and bone homeostasis [32]. Additionally, the abundance of aspartic acid in CPI (8.44 g/100 g sample) also possibly facilitated the osteogenic differentiation of the cultured cells. Aspartic acid has been shown to promote the osteogenic differentiation of hMSCs compared to glutamic acid [33]. Our results show that the presence of aliphatic amino acids, such as leucine and valine, could also increase osteoblastic differentiation. Thus, the amino acid composition of CPI is a crucial factor in determining its osteoinductiveness.

The FTIR spectroscopy analysis additionally reflects the possible secondary structure of CPI peptides [34]. The abundance of valine, isoleucine, and threonine, along with the FTIR peaks at 1,635 and 1,517 cm$^{-1}$, are correlated with the possible involvement of β-sheet conformations in the CPI. β-sheet-rich polypeptides are known to mimic the extracellular matrices for stem cell culture [35].

Despite the presence of essential amino acids and $\beta$-sheet-rich polypeptides in crude CPI, we have noted the existence of insoluble CPI fragments deposited on hBMSCs, indicating their incomplete solubility at a physiological pH. Moreover, the size and colloidal stability of the CPI were observed to determine the suitable treatment concentration (a maximum of 2% CPI in our experiment), which is mostly determined based on the isoelectric point of the CPI (pH 4.0). At a physiological pH, ~ 70% CPI solubility has been reported previously [36].

Furthermore, our experimental results also demonstrated that CPI has a comparable anti-oxidant capacity to that of ascorbate. The concentration-dependent scavenging potential of CPI is attributed to numerous factors, including the amino acid composition, the presence of different functional groups in their structure, and the presence of lower molecular weight peptide fractions in the CPI [37]. Low molecular weight peptides ($\leq$3 kDa) have been reported to show an enhanced antioxidant activity compared to larger peptides [38]. The MALDI-TOF MS spectrum of the CPI revealed a wide range of low molecular weight (below 6 kDa) peptides, which might have contributed to the reducing ability of the CPI. Hydrophobic amino acids and negatively charged amino acids (Glu and Asp) are well reported to be potential scavengers of free radicals [39,40], which explains the scavenging properties of the CPI. Additionally, the functional groups present in the antioxidants [41] also play a crucial role in determining its antioxidative property. The presence of a–C = O (carbonyl) group, as indicated by the FTIR peak at 1,635 cm$^{-1}$ of the CPI, is correlated with its antioxidative properties.

We investigated the possible cytotoxicity of CPI before determining its osteoinductive potential. We observed that the extracted CPI was highly biocompatible, as evidenced by an increased viability, a reduced cell death, an unaltered colony formation pattern, and the absence of apparent cytoskeletal/nuclear morphological damage in hBMSCs upon CPI treatment. Biocompatibility is also greatly determined by the surface morphology of the treated material [42]. The surface morphology of the undissolved CPI fraction supported cell survivability up to a 2% CPI concentration.

Next, we investigated the osteoinductive potential of CPI. We observed higher mineralization in the CPI treated group, which is also supported by the IHC scores of ALP expression. Additionally, we observed an increase in the expression of *Runx2* and *BSP* up to 14 days of CPI treatment. An increased expression of *Runx2*, *ALP*, *BSP*, and *OCN* and a decreased expression of *OSX* after 14 days of CPI treatment indicate a dynamic control of gene expression promoting osteogenic differentiation. The protein expression of Runx2, ALP, OCN, and OPN showed similar results, as evidenced by the comparable fluorescence intensities in our experiment.

Collectively, we report the successful extraction of ~ 90% of CPI from two-spotted crickets and investigated its osteoinductive properties on hBMSCs. The extracted cricket protein isolate (CPI) in our study most likely exhibited a $\beta$-sheet confirmation. The nutritional composition showed an abundance of essential amino acids in the CPI. The presence of a wide range of peptides in the CPI, as indicated by their molecular weights, reflects their possible combined role in determining stem cell fate. Notably, higher cell viability was observed in CPI-treated hBMSCs than in the control cells, showing the excellent biocompatibility of CPI. CPI also exhibited an enhanced antioxidant potential compared to the control. As evident from the ARS staining results, the increased mineralization is correlated with an increase in the *Runx2*, *ALP*, and *BSP* gene expression, confirming the osteoinductive property of CPI. Hence, the cricket-derived protein isolate is a suitable protein isolate for the osteogenic induction of hBMSCs. CPI is also a potential protein isolate that can be used as a precursor for the fabrication of scaffolds for bone regeneration. CPI can also serve as a cost-effective source of protein supplements for osteogenic differentiation. However, some aspects of this process are still under investigation. The extent to which the extraction method determined the CPI structural

and functional features is unclear. In addition, the potential of CPI to trigger multilineage differentiation is yet to be studied.

## Supporting information

**S1 Fig. MALDI TOF MS spectra of CPI for the determination of molecular weight of peptides below 6kDa.**
(DOCX)

**S2 Fig. Cell death percentage in control and 2% CPI.**
(DOCX)

**S3 Fig. Representative bright-field images of CPI-treated hBMSCs at indicated time intervals (Magnification ×20).** Arrowhead indicates the presence of CPI aggregates deposited on the surface of cells.
(DOCX)

**S4 Fig.** ROI intensity profile of (a) control and (b) 2% CPI.
(DOCX)

**S1 Table. List of antibodies used for immunofluorescence staining.**
(DOCX)

**S1 Raw images.**
(DOCX)

## Acknowledgments

Authors would like to thank the Central Laboratory of Kangwon National University, Chuncheon, Republic of Korea, for providing SEM image, FTIR data, and MALDI TOF MS spectra.

## Author Contributions

**Conceptualization:** Keya Ganguly, Sayan Deb Dutta.

**Data curation:** Keya Ganguly, Sayan Deb Dutta, Min-Soo Jeong.

**Formal analysis:** Keya Ganguly, Sayan Deb Dutta.

**Funding acquisition:** Seong-Jun Cho, Ki-Taek Lim.

**Investigation:** Keya Ganguly.

**Methodology:** Keya Ganguly, Sayan Deb Dutta, Min-Soo Jeong.

**Project administration:** Seong-Jun Cho, Ki-Taek Lim.

**Resources:** Seong-Jun Cho.

**Supervision:** Seong-Jun Cho, Ki-Taek Lim.

**Validation:** Keya Ganguly, Dinesh K. Patel.

**Visualization:** Keya Ganguly, Dinesh K. Patel, Ki-Taek Lim.

**Writing – original draft:** Keya Ganguly.

**Writing – review & editing:** Sayan Deb Dutta, Dinesh K. Patel, Ki-Taek Lim.

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
