## [Decision Letter · Decision Letter 0]

5 Jan 2021

PONE-D-20-38399

Naturally-derived protein biomaterials from Gryllus bimaculatus for improved antioxidant and stem cell differentiation

PLOS ONE

Dear Dr. Lim,

Thank you for submitting your manuscript to PLOS ONE. After careful consideration, we feel that it has merit but does not fully meet PLOS ONE’s publication criteria as it currently stands. Therefore, we invite you to submit a revised version of the manuscript that addresses the points raised during the review process.

Although of some interest, this manuscript and study needs to be considerably improved since the Title.

The pitfalls are numerous and include some methodological mistakes.

The Authors must follow all the criticisms raised by the two referees and amend the manuscript as requested.

We look forward to receiving your revised manuscript.

Kind regards,

Gianpaolo Papaccio, M.D., Ph.D.

Academic Editor

PLOS ONE

Journal Requirements:

Reviewers' comments:

Reviewer's Responses to Questions

**Comments to the Author**

1. Is the manuscript technically sound, and do the data support the conclusions?

Reviewer #1: Partly

Reviewer #2: Partly

2. Has the statistical analysis been performed appropriately and rigorously? 

Reviewer #1: Yes

Reviewer #2: No

3. Have the authors made all data underlying the findings in their manuscript fully available?

Reviewer #1: Yes

Reviewer #2: Yes

4. Is the manuscript presented in an intelligible fashion and written in standard English?

Reviewer #1: No

Reviewer #2: Yes

5. Review Comments to the Author

Reviewer #1: The authors aimed to investigate the effects of CPI on the osteogenic differentiation of hBMSCs.

The manuscript is interesting, but there are some points that must be clarified.

I suggest to improve the title. First of all, it is badly written in English. I suggest to use the word biopolymer instead of biomaterial. Biopolymers are polymers produced by living organisms. Cellulose, proteins, peptides, are all examples of biopolymers used in tissue engineering. For example, I suggest a possible title as following: "Naturally derived proteins extract from Gryllus bimaculatus improves antioxidant properties and promotes osteogenic differentiation of hBMSCs".

In figure 5a, the authors must report what the arrows indicate in the pictures.

The osteogenic differentiation must be evaluated also at 21 days .

The immunofluorescence showed in figure 8 (in particular for 2%CPI) is very bad. Dye deposits are detectable and the staining is aspecific. The Authors indicate these deposits as presence of CPI on surface of hBMSCs. How can the authors confirm this? If this is, it means that the CPI is not soluble. Incomplete solubility makes CPI difficult to use. Please, clarify better this point.

ALP must be performed using histological staining.

Moreover, the Authors stated that: “no significant difference was noted in the mean intensities among the control, and the CPI treated cells”.

In figure 8, the statistic is indicated using asterisks. This point must be clarified and reported in Results.

The Discussion must be revised, it is too long and dispersive.

Reviewer #2: 518 / 5000

Risultati della traduzione

Although the study was previously not sufficiently suitable for publication on Plos One, the experiments carried out by the authors following the suggestions of the reviewers, have significantly improved the paper.

Despite this, some concerns need to be addressed.

First of all, the title should be rewritten: it should be improved grammatically, with a verb that would make the meaning of the paper clearer.

Osteogenesis should also be evaluated at 21 days.

The discussion needs to be shortened.

6. PLOS authors have the option to publish the peer review history of their article (what does this mean?). If published, this will include your full peer review and any attached files.

Reviewer #1: No

Reviewer #2: No

---

## [Author Response · Author response to Decision Letter 0]

26 Feb 2021

Response to Reviewers:

Reviewer 1:

The authors aimed to investigate the effects of CPI on the osteogenic differentiation of hBMSCs. The manuscript is interesting, but there are some points that must be clarified.

Author’s response: Thank you very much for reviewing our manuscript. We have made the necessary corrections based on your valuable suggestion. 

Comment #1. I suggest to improve the title. First of all, it is badly written in English. I suggest to use the word biopolymer instead of biomaterial. Biopolymers are polymers produced by living organisms. Cellulose, proteins, peptides, are all examples of biopolymers used in tissue engineering. For example, I suggest a possible title as following: "Naturally derived proteins extract from Gryllus bimaculatus improves antioxidant properties and promotes osteogenic differentiation of hBMSCs".

Author’s response: Thank you for your kind recommendation. We agree with your suggested title and have updated the same in our revised manuscript. 

[Naturally-derived protein extract from Gryllus bimaculatus improves antioxidant properties and promotes osteogenic differentiation of hBMSCs].

Comment #2. In figure 5a, the authors must report what the arrows indicate in the pictures.

Author’s response: Thank you for your comment. We have indicated the necessary information in the revised manuscript.

[Fig. 5. In vitro cytotoxicity evaluation of CPI on hBMSCs after 3 days of incubation. (a) optical microscopy images after Giemsa staining (b) fluorescence microscopy images of F-actin. The white arrows indicate the presence of stained CPI on the surface of hBMSCs].

Comment #3. The osteogenic differentiation must be evaluated also at 21 days.

Author’s response: Thank you very much for your suggestion. We have included the hBMSC differentiation using the ARS staining procedure (Fig. 5.) and immunohistochemistry of ALP expression (Fig. 6.) for 7, 14, and 21 days in our revised manuscript.

Comment #4. The immunofluorescence showed in figure 8 (in particular for 2%CPI) is very bad. Dye deposits are detectable and the staining is aspecific. The Authors indicate these deposits as presence of CPI on surface of hBMSCs. How can the authors confirm this? If this is, it means that the CPI is not soluble. Incomplete solubility makes CPI difficult to use. Please, clarify better this point.

Author’s response: Thank you very much for deeply reviewing our manuscript. We agree with your opinion. The CPI used in this study is partially soluble in DMEM at the physiological pH and hence some of the aggregated structure are visible on the surface of the cultured hBMSCs. To confirm the aggregates as CPI, we have taken the bright field images of the cells after desired treatment and is indicated in Supplementary Fig. 3 of the revised manuscript. Thus, the fluorescence microscopy images taken at different time period also indicates the presence of CPI aggregates on the surface of cells as shown in Fig. 8. We completely agree that soluble protein is more convenient for cellular interaction or metabolism. However, the purpose of our study was to check the potential of crude CPI on hBMSCs osteogenic potential at physiological pH. The size and the colloidal stability of the up to 2% CPI at physiological pH was found to favor hBMSCs growth and differentiation. Based on our study, we conclude that the use of up to 2% crude CPI is favorable for hBMSCs growth.

Comment #5. ALP must be performed using histological staining. Moreover, the Authors stated that: “no significant difference was noted in the mean intensities among the control, and the CPI treated cells”.

Author’s response: Thank you very much for your suggestion. We have included the immunohistochemistry staining of ALP after 7, 14, and 21 days of CPI treatment. As the protein expression for ALP was performed using the immunocytochemistry analysis, we could confirm the successful expression of the protein in both the treated and the control cells, indicating the CPI treatment favored the expression of the osteogenic marker proteins similar to control cells. 

Comment #6. In figure 8, the statistic is indicated using asterisks. This point must be clarified and reported in Results.

Author’s response: Thank you for your kind information. We have discussed this point in the “Results” section in our revised manuscript. 

Comment #7. The Discussion must be revised, it is too long and dispersive.

Author’s response: Thank you for your suggestion. We have updated the discussion in our revised manuscript as per your recommendation. 

Reviewer 2: 

Risultati della traduzione

Although the study was previously not sufficiently suitable for publication on Plos One, the experiments carried out by the authors following the suggestions of the reviewers, have significantly improved the paper. Despite this, some concerns need to be addressed.

Comment #1. First of all, the title should be rewritten: it should be improved grammatically, with a verb that would make the meaning of the paper clearer.

Author’s response: Thank you very much for your valuable comment. We have changed the title in the revised manuscript as follows:

[Naturally-derived protein extract from Gryllus bimaculatus improves antioxidant properties and promotes osteogenic differentiation of hBMSCs].

Comment #2. Osteogenesis should also be evaluated at 21 days.

Author’s response: Thank you very much for your comment. We have evaluated the osteogenesis through ARS staining procedure for hBMSCs mineralization quantification (Fig. 5.) and ALP expression through immunohistochemistry (Fig. 6.) for 7, 14, and 21 days in the revised manuscript.

Comment #3. The discussion needs to be shortened.

Author’s response: Thank you very much for your kind recommendation. Based on your suggestion, we have shortened the discussion section in the revised manuscript.

---

## [Decision Letter · Decision Letter 1]

16 Mar 2021

Naturally-derived protein extract from Gryllus bimaculatus improves antioxidant properties and promotes osteogenic differentiation of hBMSCs

PONE-D-20-38399R1

Dear Dr. Lim,

We’re pleased to inform you that your manuscript has been judged scientifically suitable for publication and will be formally accepted for publication once it meets all outstanding technical requirements.

Kind regards,

Gianpaolo Papaccio, M.D., Ph.D.

Academic Editor

PLOS ONE

Additional Editor Comments (optional):

Reviewers' comments:

Reviewer's Responses to Questions

**Comments to the Author**

1. If the authors have adequately addressed your comments raised in a previous round of review and you feel that this manuscript is now acceptable for publication, you may indicate that here to bypass the “Comments to the Author” section, enter your conflict of interest statement in the “Confidential to Editor” section, and submit your "Accept" recommendation.

Reviewer #1: All comments have been addressed

Reviewer #2: All comments have been addressed

2. Is the manuscript technically sound, and do the data support the conclusions?

Reviewer #1: Yes

Reviewer #2: (No Response)

3. Has the statistical analysis been performed appropriately and rigorously? 

Reviewer #1: Yes

Reviewer #2: (No Response)

4. Have the authors made all data underlying the findings in their manuscript fully available?

Reviewer #1: Yes

Reviewer #2: (No Response)

5. Is the manuscript presented in an intelligible fashion and written in standard English?

Reviewer #1: Yes

Reviewer #2: (No Response)

6. Review Comments to the Author

Reviewer #1: (No Response)

Reviewer #2: (No Response)

7. PLOS authors have the option to publish the peer review history of their article (what does this mean?). If published, this will include your full peer review and any attached files.

Reviewer #1: No

Reviewer #2: No

---

## [Editor Report · Acceptance letter]

21 May 2021

PONE-D-20-38399R1 

Naturally-derived protein extract from *Gryllus bimaculatus* improves antioxidant properties and promotes osteogenic differentiation of hBMSCs 

Dear Dr. Lim:

I'm pleased to inform you that your manuscript has been deemed suitable for publication in PLOS ONE. Congratulations! Your manuscript is now with our production department. 

Kind regards, 

on behalf of

Prof. Gianpaolo Papaccio 

Academic Editor

PLOS ONE